# Does route matter? Impact of route of oxytocin administration on postpartum bleeding: A double-blind, randomized controlled trial

Jill Durocher[1]*, Ilana G. Dzuba[1], Guillermo Carroli[2], Elba Mirta Morales[3], Jesus Daniel Aguirre[3], Roxanne Martin[1], Jesica Esquivel[3], Berenise Carroli[2], Beverly Winikoff[1]

**1** Gynuity Health Projects, New York, New York, United States of America, **2** Centro Rosarino de Estudios Perinatales (CREP), Rosario, Argentina, **3** Hospital Materno Neonatal E.T. de Vidal, Corrientes, Argentina

* jdurocher@gynuity.org

**Data Availability Statement:** The dataset used for this analysis is available at Harvard Dataverse

## Abstract

### Objective

We assessed the impact of intravenous (IV) infusion versus intramuscular (IM) oxytocin on postpartum blood loss and rates of postpartum hemorrhage (PPH) when administered during the third stage of labor. While oxytocin is recommended for prevention of PPH, few double-blind studies have compared outcomes by routes of administration.

### Methods

A double-blind, placebo-controlled randomized trial was conducted at a hospital in Argentina. Participants were assigned to receive 10 IU oxytocin via IV infusion or IM injection and a matching saline ampoule for the other route after vaginal birth. Blood loss was measured using a calibrated receptacle for a 1-hour minimum. Shock index (SI) was also calculated, based on vital signs measurements, and additional interventions were recorded. Primary outcomes included: the frequency of blood loss ≥500ml and mean blood loss.

### Results

239 (IV infusion) and 241 (IM) women were enrolled with comparable baseline characteristics. Mean blood loss was 43ml less in the IV infusion group (p = 0.161). Rates of blood loss ≥500ml were similar (IV infusion = 21%; IM = 24%, p = 0.362). Women in the IV infusion group received significantly fewer additional uterotonics (5%), than women in the IM group (12%, p = 0.007). Women with PPH in the IM group experienced a larger increase in SI after delivery, which may have influenced recourse to additional interventions.

### Conclusions

The route of oxytocin administration for PPH prevention did not significantly impact measured blood loss after vaginal birth. However, differences were observed in recourse to

Network: https://dataverse.harvard.edu/dataset.xhtml?persistentId=doi:10.7910/DVN/MDZRKU.

**Funding:** This study was funded by The Bill & Melinda Gates Foundation (grant ID OPPGH5295). The funder had no role in the study design, data collection and analysis, decision to publish, or preparation of the manuscript.

**Competing interests:** The authors have declared that no competing interests exist.

additional uterotonics, favoring IV infusion over IM. In settings where IV lines are routinely placed, oxytocin infusion may be preferable to IM injection.

## Introduction

Postpartum hemorrhage (PPH) is a major cause of maternal death and morbidity worldwide and is most commonly a result of uterine atony [1,2]. The administration of oxytocin during the third stage of labor is widely promoted to support uterine contractions after childbirth and reduce the risk of bleeding [3]. Comparative studies of oxytocin either in combination with other components of active management or alone show that it is safe and effective in reducing PPH [4]. While obstetric guidelines universally recommend oxytocin as the uterotonic of choice to prevent PPH, questions remain regarding the optimal route for its administration. Indeed, a comparison of guidelines, released by professional associations and leading health authorities, reveals significant heterogeneity in the recommendations for administering prophylactic oxytocin [5]. For instance, some guidelines specify only intramuscular (IM) administration for prevention of PPH [6–8], whereas others, including the World Health Organization, recommend both IM injection and intravenous (IV) routes as equal alternatives to deliver oxytocin during the third stage of labor [3,9].

The administration of oxytocin via IM injection has become increasingly widespread due to its favorable safety profile and the practical advantages over IV administration, particularly in lower levels of care where IV placement may be less feasible. Yet, evidence also supports IV administration, and some researchers report improved clinical outcomes that favor IV routes over IM [10–13], including the authors of a large systematic review of comparative studies on prophylactic oxytocin [4]. An earlier pharmacokinetic investigation, published in 1972, on circulating oxytocin levels, also found that plasma levels rose more rapidly following IV administration in the third stage of labor and reached a higher peak than after IM [14]. Importantly though, a 2018 Cochrane review identified a lack of randomized controlled trials that compared the safety and effectiveness of prophylactic oxytocin when administered via different routes [15]. It is also noted that relatively few studies of prophylactic oxytocin have been blinded [4]. The authors of both systematic reviews called for more high quality research to determine if route of administration matters [4,15].

To address this gap in the evidence, we evaluated the effect of route of prophylactic administration of oxytocin on PPH outcomes, and, more specifically, whether IV infusion results in less postpartum blood loss than IM injection.

## Materials and methods

A double-blind, placebo-controlled randomized trial was conducted to compare IV infusion and IM injection of 10 IU oxytocin during the third stage of labor after vaginal delivery to assess the impact of route on postpartum blood loss and rates of PPH. A local ethics committee (Centro Rosarino de Estudios Perinatales) in Argentina approved the study prior to its implementation in a public, tertiary-level hospital in Corrientes, Argentina (Hospital Materno Neonatal E.T. de Vidal). The trial was registered with Clinical trials.gov (NCT02954068) on 3 Nov. 2016 (URL: https://clinicaltrials.gov/ct2/show/NCT02954068).

### Participants and enrollment

All women presenting in active labor with a live fetus at the study hospital were invited to participate in the study and screened for eligibility. Women were ineligible if they had a cesarean

delivery or were unable to give informed consent for any reason. There were no other exclusion criteria based on risk factors or delivery complications. Women were asked to confirm their willingness to respond to a few questions about their pregnancy history, have their hemoglobin measured with a Hemocue® (Angelholm, Sweden) device, and have an IV line in place at delivery (standard at this hospital). Participants' background characteristics and baseline assessments prior to delivery were documented only after written informed consent was obtained. Data collection was undertaken by trained maternity ward staff.

## Study interventions, randomization, and blinding

Ampoules of oxytocin and matching placebo (saline solution) were supplied for the study by a local pharmaceutical company (Instituto Biologico Argentino S.A.I.C., CABA, Argentina). Consecutively numbered study packets were prepared to maintain blinding; each packet included two identical ampoules–one 10 IU oxytocin ampoule and a matching placebo ampoule, one of which was labeled for IV infusion and the other for IM administration in accordance with the randomization scheme. The computer-generated randomization code in blocks of ten was created by Gynuity Health Projects in New York and not revealed until after data collection and database cleaning were completed. The study packets were labeled with a unique identification number, placed in sequential order in a dispenser, allowing individual extraction, and kept in a locked refrigerator per recommended storage requirements for oxytocin to maintain its quality and effect. The randomization code was not shared with hospital staff or local investigators, and regular monitoring ensured that packets were used in sequential order.

When birth was imminent, the next packet was removed from the dispenser and the woman was randomized to the study group. Staff was trained to remove both ampoules and prepare them for immediate administration after the birth of the baby per the labeled route on each ampoule. Because medicines given via IM injection are absorbed more slowly, staff was trained to administer the IM ampoule first, usually in the thigh. The second ampoule labeled for IV infusion was mixed in a 500cc bag of saline solution and initiated immediately after the IM injection and administered with an 18–20 gauge needle at a rate of approximately 12cc per minute for 40 minutes.

## Post-randomization study procedures

Immediately after the birth of the baby, postpartum blood loss was measured using a polyurethane receptacle (Brasss-V Drapes®, Excellent Fixable Drapes, India) with calibrated markings in increments of 100cc. Blood, along with any clots, was funneled into the drape for a minimum of 1 hour or until active bleeding ceased. Pulse and blood pressure were measured at 15-minute intervals for one hour postpartum, using an automated blood pressure instrument (Omron BP785, Omron Healthcare Co., Ltd.).Women who were diagnosed with PPH were treated according to standard hospital protocol. Blood loss was measured at the time of PPH diagnosis and again when active bleeding stopped. Irrespective of PPH diagnosis, blood loss amounts were systematically recorded for all women at 30 and 60 minutes postpartum.

All actions taken to manage bleeding were documented, and the woman's general condition was noted prior to discharge. A second hemoglobin assessment was conducted and documented 24–48 hours postpartum, and when possible, at least 12 hours after removal of any IV line.

## Study outcomes and sample size

The primary outcomes were the proportion of women with a blood loss of 500ml or greater and mean total blood loss, based on objective measurement. Secondary outcomes included: average blood loss at 30 and 60 minutes postpartum, average vital sign measures, including

shock index, the rate of severe PPH ≥1000ml, administration of additional uterotonics and/or other interventions, change in hemoglobin from pre- to post-delivery, and any adverse effects reported. Shock index (SI), defined as the ratio of pulse to systolic blood pressure, was analyzed at study completion and interpreted based on normal (SI<0.90) and abnormal (SI≥0.90 values described in the literature for obstetric populations [16–17].

Prior to this study, research examined the effect of oxytocin route on postpartum blood loss [10–11], including a secondary analysis that explored the effectiveness of active management interventions on postpartum blood loss when used in combination or alone. That analysis found that IV administration of oxytocin alone (i.e. in the absence of other active management components performed), compared to IM oxytocin alone, was associated with a 76% reduction in the rate of PPH (p<0.001) [11]. However, when IV or IM oxytocin was given during the third stage of labor, in combination with other interventions, such as controlled cord traction, the effect of route became negligible. Another small randomized trial was conducted to directly compare the effect of oxytocin infusion vs. IM injection on measured blood loss following vaginal deliveries that had other components of the active management of the third stage of labor performed and documented a statistically significant reduction in measured blood loss ≥ 500ml (IV infusion: 9% (15/161) and IM: 20% (32/161); RR 0.47 95% CI 0.25–0.85) [11]. Based on this evidence[10–11], we hypothesized that administration of oxytocin via IV infusion would result in a 50% lower rate of PPH (defined as blood loss ≥500ml) than IM administration and a difference of 50ml when comparing average total blood loss between study groups. According to available information from the site on their PPH rate and knowledge that PPH rates are highly variable [18] and are influenced by other practices during the third stage of labor [10], we selected an 18% PPH ≥500ml rate for sample size calculation. Assuming that IV infusion would be associated with 50% less PPH than IM injection, 442 randomized cases (221 per group) were required, based on alpha = 0.05, power = 0.80, and a two-tailed test. The sample was increased by 10% to account for any loss to follow-up or missing data that would impact the assessment of the main outcomes, resulting in a minimum enrollment of 486 women. This sample size was also sufficient to detect a difference in mean total blood loss of 50ml or greater in the two study groups.

## Statistical methods

All study forms were translated into Spanish, and data from completed study forms were entered into a Microsoft Access database (version 2007). Data were then exported to SPSS 19.0 (IBM, Chicago, IL, USA) for analysis, using an intention-to-treat approach. Descriptive statistics for categorical variables were summarized as numbers and percentages and for continuous variables as means (standard deviation). Medians (interquartile ranges) were calculated for continuous variables that were determined not to be normally distributed based on skewness and kurtosis z-values (<-1.96 or >+1.96) and a Shapiro-Wilk's test (p<0.05). Group differences for outcomes based on categorical variables were assessed using Pearson $\chi^2$ or Fisher's exact tests (as appropriate). Continuous variables were analyzed using Independent t-tests, or if not normally distributed, using non-parametric tests (Mann-Whitney U). The relative risk and risk difference (and 95% confidence intervals) were calculated for the main study outcomes, as appropriate. Statistical significance was defined as p-value <0.05. Analyses were not adjusted for any confounders.

## Results

From Dec. 2016 to Sept. 2017, 543 women presenting in labor at the study hospital were screened for eligibility and gave consent to participate (Fig 1). Following enrollment, 60

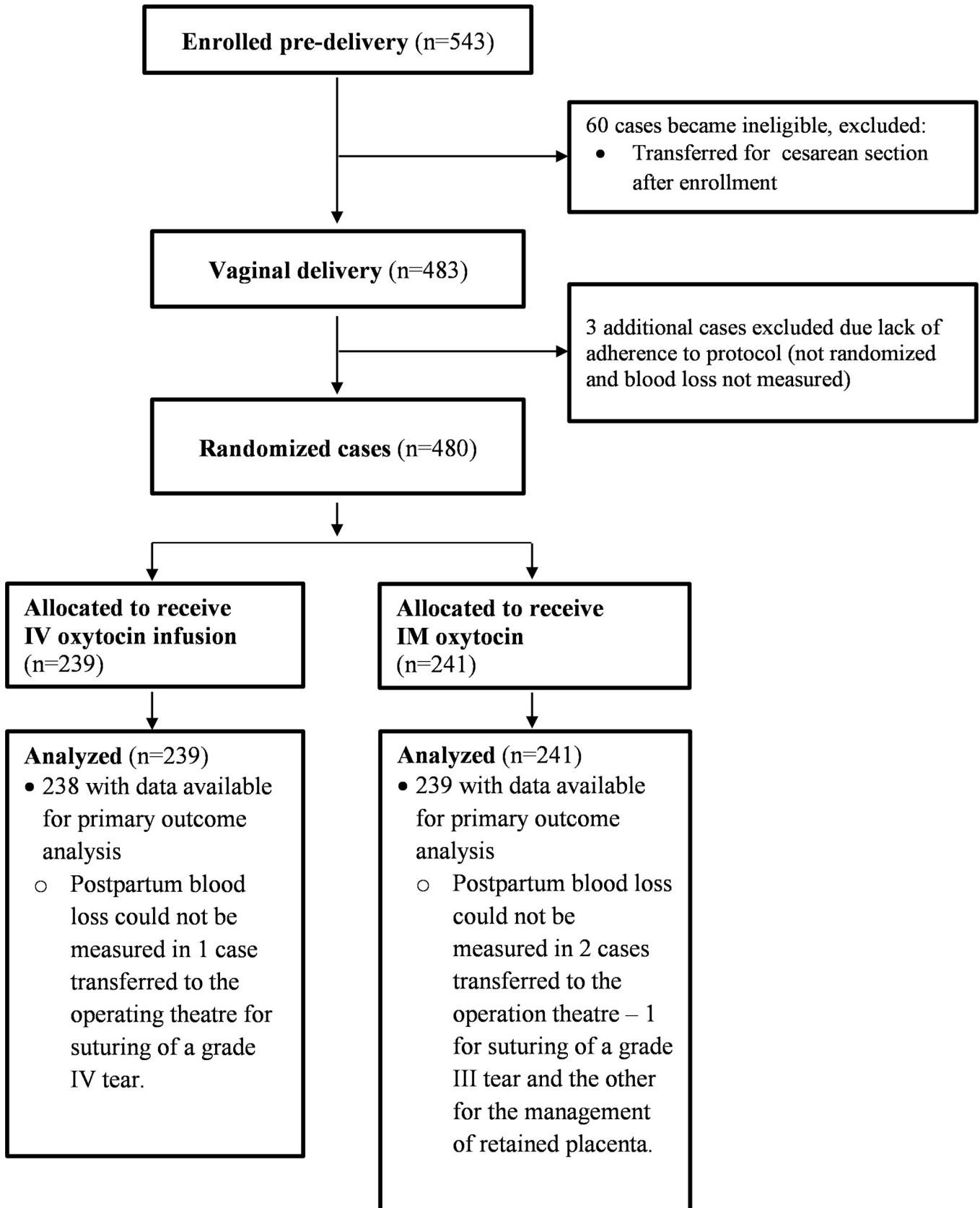

**Fig 1. Consort flow diagram: Trial profile.**

women were transferred for cesarean delivery and were withdrawn from the study. Three additional cases were excluded prior to randomization due to staff error and missing post-delivery measurements. In total, 480 women who delivered vaginally were randomized to receive 10 IU oxytocin as prophylaxis against PPH either intravenously or intramuscularly and had adequate data for analysis. The IV infusion and IM study groups consisted of 239 and 241 women, respectively, and had comparable baseline characteristics at enrollment (Table 1).

All randomized participants were administered both study ampoules from the study packets per the labeled routes. The IM-labeled ampoule was given within 1 minute of the birth of the baby for nearly all women (93%) in both study arms (Table 2). The average time to complete the IV infusion was 40 minutes in both randomized groups. Difficulties administering the IV infusion occurred in two cases (one in each study group)–one was an infiltrated line and the other was a dislodged IV during delivery. There were no reports of problems occurring with the IM injection. Rates of controlled cord traction and uterine massage were similar among study groups (Table 2).

Objective blood loss measurement revealed similar rates of PPH ≥500ml in both study arms (IV infusion: 20.6% and IM: 23.8%, p = 0.392) (Table 3). Although mean blood loss was 42ml less in the IV infusion group, compared to the IM group, this difference was not statistically significant (p = 0.161). Median total blood loss was confirmed to be 300ml in each study arm. Average blood loss (recordings at 30 min and at 60 min postpartum), as well as systolic and diastolic blood pressure and pulse readings were also similar in the two groups at all times measured.

Shock index (SI) was also analyzed at 15-minute intervals postpartum. Fig 2 shows that women without PPH (n = 375) had relatively stable SI levels with a median of 0.74 pre-delivery and median SIs of 0.78, 0.75, 0.75, and 0.74 at 15, 30, 45, and 60-minutes postpartum, respectively, without any differences noted between study arms. Among those who developed PPH

**Table 1. Baseline characteristics by study group.**

| Baseline characteristics | IV infusion group (n = 239) | IM group (n = 241) |
|---|---|---|
| Woman's age, mean ± SD | 24.1 ± 5.2 | 24.3 ± 5.7 |
| Education level, n (%) | | |
| None | 4 (1.7) | 0 (0) |
| Primary | 81 (33.9) | 87 (36.1) |
| Secondary or higher | 154 (64.4) | 154 (63.9) |
| # of pregnancies, mean ± SD | 2.3 ± 1.5 | 2.4 ± 1.8 |
| Primigravida, n (%) | 93 (38.9) | 92 (38.2) |
| Gestational age, mean ± SD | 38.6 ± 2.1 | 38.9 ± 1.9 |
| Had previous PPH, n (%) | 2 (0.8) | 4 (1.7) |
| Hb pre-partum, mean ± SD [a] | 11.8 ± 1.5 | 11.7 ± 1.5 |
| Pre-delivery Hb < 11.0 g/dL, n (%) [a] | 61 (25.6) | 68 (28.2) |
| Pre-delivery shock index, mean ± SD | 0.75 ± 0.2 | 0.75 ± 0.2 |
| Labor induced with uterotonics, n (%) | 16 (6.7) | 25 (10.4) |
| Labor augmented with uterotonics, n (%) | 40 (16.7) | 27 (11.2) |
| Singleton, n (%) | 237 (99.2) | 238 (98.8) |
| Episiotomy, n (%) | 104 (43.5) | 111 (46.1) |

[a] N = 238 in the IV infusion group– 1 case had missing data on pre-partum Hb.

**Table 2. Active management of the third stage of labor by study group.**

| Delivery characteristics [a] | IV infusion group (n = 239) | IM group (n = 241) |
|---|---|---|
| Prophylaxis initiated within 1 minute of birth of baby, n (%) [b] | 221 (92.5) | 225 (93.8) |
| IV infusion time (min.) [c] | | |
| mean ± SD | 40.1 ± 8.9 | 40.3 ± 9.7 |
| (range) | (10–100) | (15–90) |
| Controlled cord traction, n (%) | 229 (95.8) | 237 (98.3) |
| Uterine massage, n (%) | 91 (38.1) | 81 (33.6) |
| Time (min.) to placental delivery [d] | | |
| mean ± SD | 6.4 ± 5.7 | 6.6 ± 7.2 |
| (range) | (0–40) | (0–62) |

[a] No statistically significant differences measured; p>0.05 for all variables.

[b] N = 240 in the IM group– 1 case missing data on time of IM injection administration.

[c] N = 236 in the IV infusion group and N = 240 in the IM group due to missing data on time of infusion completion.

[d] N = 239 in the IM group– 2 cases missing data on time of placental expulsion.

(Fig 2), median SI values were statistically significantly higher at each interval during the first hour postpartum, compared to women without PPH. This pattern was consistent irrespective of the study group. Median SI values at all time points, including changes in SI between the intervals, were comparable between study groups among PPH and non-PPH cohorts with one exception (S1 Fig). Women in the IM group who developed PPH experienced a sharp rise in SI from pre-delivery to 15-minutes postpartum (median Δ 0.20, IQR 0.08, 0.33), which was higher than in the IV infusion group (median Δ 0.13 IQR -0.02, 0.22, p = 0.048). Overall, the proportion of women with an elevated SI ≥ 0.9 at 15-minutes postpartum was higher in the IM group (80/241; (33%), compared to 25% in the IV infusion group (60/239) (p = 0.051) (S2 Fig).

While blood loss outcomes were not clinically or statistically different between study groups, women in the IV infusion group received significantly fewer additional uterotonics (5%), compared to women randomized to IM injection (12%, p = 0.007) (Table 3). Also, fewer women received other interventions in the IV infusion group, including exploration under anesthesia, administration of plasma expanders, and blood transfusion, though none were statistically significant (Table 3). The median amount of IV fluids infused over the course of delivery was 1500 ml in both arms, and postpartum hemoglobin outcomes were similar between study groups (Table 3).

A closer examination of PPH cases in this study (n = 105) confirms that uterine atony (41.9%) and episiotomy (34.3%) were the most frequently reported causes of PPH; their occurrence was similar across study arms. All women made a full recovery from their bleeding, and no maternal deaths or severe outcomes occurred in the entire study population. There were also no reports of any adverse effects associated with prophylactic oxytocin administration in either study group.

## Discussion

This double-blind, randomized controlled trial found that rates of PPH and the average volume of total blood loss were similar whether oxytocin (10 IU) was given during the third stage of labor by IV infusion or IM injection. While there were no differences in the study's two primary outcomes based on blood measurement, the infusion of prophylactic oxytocin intravenously was associated with a clinically and statistically significant reduction in the use of

**Table 3. Trial outcomes by study group.**

| Outcomes | IV infusion group (n = 239) | IM group (n = 241) | Relative Risk / Estimate (95% CI) | p-value |
|---|---|---|---|---|
| *Primary outcomes* | | | | |
| Blood loss $\geq$ 500 ml, n (%) [a] | 49 (20.6) | 57 (23.8) | 0.86 (0.62, 1.21) | 0.392 |
| Total blood loss, mean ± SD [a] | 364 ± 323 | 406 ± 344 | -42.8 ml (-102.8, 17.2) | 0.161 |
| *PPH rates* | | | | |
| PPH diagnosed, n (%) | 48 (20.1) | 57 (23.7) | 0.85 (0.61, 1.19) | 0.344 |
| Time (min.) to diagnosis [b] | | | | |
| median (IQR) | 30.0 (15, 41) | 25.0 (15, 30) | | 0.174 |
| Primary cause due to atony, n (%) | 19 (39.6) | 25 (43.9) | 0.90 (0.57, 1.43) | 0.658 |
| Blood loss $\geq$ 750 ml, n (%) [a] | 21 (8.8) | 31 (13.0) | 0.68 (0.40, 1.15) | 0.146 |
| Blood loss $\geq$ 1000 ml, n (%) [a] | 14 (5.9) | 18 (7.5) | 0.78 (0.40, 1.53) | 0.472 |
| *Additional interventions* | | | | |
| Suturing and/or tear repair, n (%) | 92 (38.5) | 100 (41.5) | 0.93 (0.75, 1.16) | 0.502 |
| Manual removal of placenta, n (%) | 0 (0) | 3 (1.2) | Cannot estimate | 0.248 |
| Bimanual compression, n (%) | 0 (0) | 3 (1.2) | Cannot estimate | 0.248 |
| Exploration under anesthesia, n (%) | 1 (0.4) | 5 (2.1) | 0.20 (0.02, 1.71) | 0.216 |
| Additional uterotonics, n (%) [c] | 13 (5.4) | 30 (12.4) | 0.44 (0.23, 0.82) | 0.007 |
| Oxytocin IV (10–20 IU), n [c] | 12 | 28 | | |
| Ergonovine IM (0.2–0.8), n [c] | 3 | 5 | | |
| Blood transfusion, n (%) | 4 (1.7) | 6 (2.5) | 0.67 (0.19, 2.35) | 0.751 |
| Plasma expanders, n (%) | 7 (2.9) | 11 (4.6) | 0.64 (0.25, 1.63) | 0.346 |
| Hysterectomy or other surgery [d], n (%) | 0 (0) | 1 (0.4) | Cannot estimate | 1.00 |
| *Hemoglobin outcomes* [e] | | | | |
| Hb post-partum, median (IQR) [b] | 10.5 (9.6, 11.2) | 10.5 (9.6, 11.2) | | 0.453 |
| (range) | (5.9–14.7) | (5.5–15.4) | | |
| Hb drop $\geq$ 2g/dL or given blood transfusion, n (%) [e] | 79 (33.5) | 74 (30.8) | 1.09 (0.84, 1.41) | 0.537 |

[a] Analysis of outcomes based on blood loss excludes three women whose measurement of blood was discontinued when transferred to the operating theatre for additional care (1 woman in the IV infusion group received suturing for grade IV tear; in the IM group, 1 woman received suturing for grade III tear and 1 woman was transferred for management of retained placenta). Baseline characteristics for these three women were not different from other cases included in analysis.

[b] Mann-Whitney U tests were calculated for secondary outcomes that were not normally distributed, including time to PPH diagnosis from delivery of baby and postpartum Hb.

[c] Administration of both uterotonics listed occurred in two women in the IV infusion group and three in the IM group.

[d] Other surgery refers to 1 woman in the IM group who received curettage for uterine evacuation due to incomplete placental expulsion.

[e] N = 236 in the IV infusion group and N = 240 in IM group due to missing postpartum Hb outcomes.

additional uterotonics when compared to its administration via IM injection (IV: 5% vs. IM: 12%; RR 0.44 95% CI 0.22–0.85). The provision of other interventions was also lower in the IV infusion group, compared to IM injection, although none were found to be statistically significant except for recourse to therapeutic uterotonics (Table 3).

An analysis of bleeding patterns by study group provides insight into reasons for less frequent use of additional uterotonics in the IV infusion group than the IM group. For instance, in the IV infusion group, median blood loss at 30 minutes postpartum was 50ml less than among women randomized to IM injection. Although this finding lacks statistical significance, it suggests that women in the IM group bled more initially after childbirth. This trend is further supported by results from a post-hoc cohort analysis of atonic PPH cases showing that despite comparable blood loss at the time of PPH diagnosis, women in the IM group were diagnosed earlier (median: 18 min) after delivery, compared to women who received oxytocin

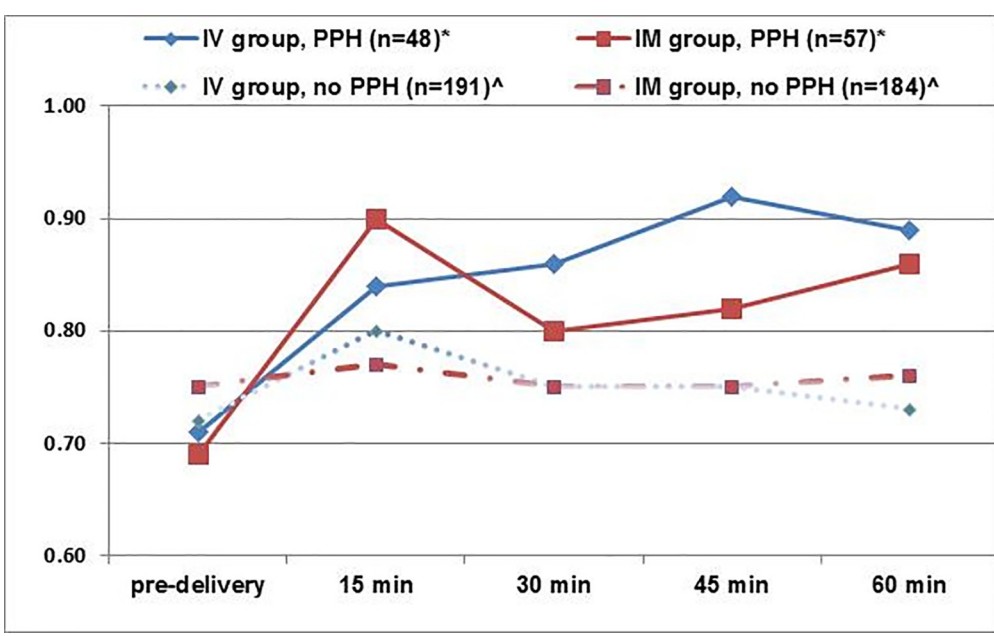

**Fig 2. Median shock index values pre-delivery and during the first hour postpartum for PPH cases and non-PPH cases by study group.** *Among PPH cases, median SIs were not statistically different between study groups at any time postpartum (p>0.05). ^Among non-PPH cases, median SIs were not statistically different between study groups at any time postpartum (p>0.05).

via IV infusion (median: 30 min, p = 0.011). These results may be explained by the slower absorption and onset of action of IM oxytocin (i.e. 3–7 minutes) versus intravenously, with an almost immediate effect [3,19].

A similar effect was also reflected in an analysis of shock index (SI) levels by study group, particularly among women diagnosed with PPH, independent of cause. Concordant with the literature [20], the PPH cases in the present study had higher SI values in than women without PPH (Fig 2). Previous studies have shown that SI increases when a woman's vital signs are compromised from bleeding, and when a rise in the pulse rate and a decline in systolic BP occur [16,17,21,22]. In Fig 2, a sharper rise in SI is seen among PPH cases in the IM group from before delivery to 15-minutes postpartum, compared to the IV infusion group (p = 0.048). This finding reinforces the evidence that women with PPH in the IM group likely experienced more blood loss after delivery than women in the IV infusion group.

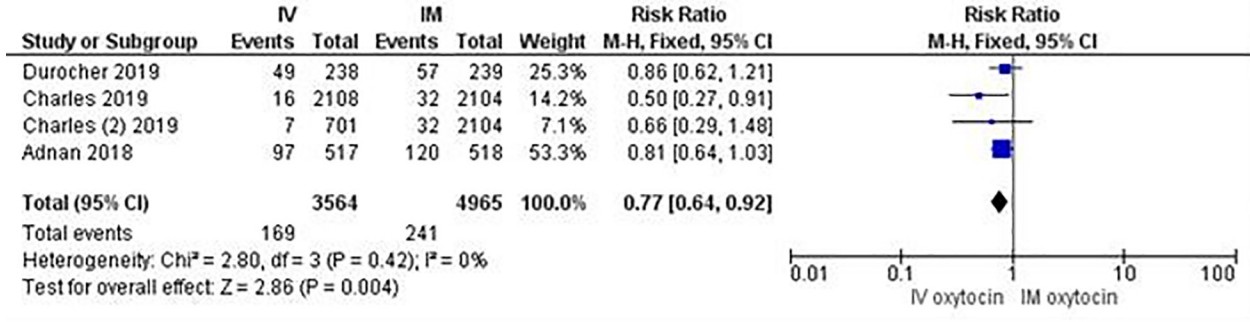

**Fig 3. Meta-analysis of three randomized controlled trials comparing IV and IM routes of oxytocin administration for PPH prevention (Outcome: Measured blood loss ≥500ml).**

**Fig 4. Meta-analysis of three randomized controlled trials comparing IV and IM routes of oxytocin administration for PPH prevention (Outcome: Measured blood loss ≥1000ml).**

Notably, contemporaneous findings from two large randomized controlled trials found PPH rates, based on objective measurement, to be significantly reduced when oxytocin prophylaxis was given via IV routes compared to IM injection after vaginal delivery [12–13]. The study by Charles et al, a three-arm trial conducted in Egypt, documented a significantly lowered risk of PPH (≥500mL) following administration of 10 IU prophylactic oxytocin via IV infusion, compared to IM injection (Relative Risk of 0.50, 95% CI 0.27–0.91), but did not confirm any difference in PPH rates when comparing IV bolus to IM injection [13]. Another large trial conducted in Ireland confirmed a significantly lower rate of severe PPH ≥1000mL following its IV bolus administration vs. IM injection (Odds Ratio of 0.54 95% CI 0.32–0.91) [12].

Both the Adnan and Charles studies had sample sizes that were more than twice the size of our study, which may have enabled the detection of a statistical difference in PPH rates between IV and IM routes [12–13]. It is also plausible that the faster infusion rate of IV oxytocin in the Egypt study or that administration in bolus improved outcomes in the IV groups of these trials. Our study had a mean IV infusion time of 40 minutes, whereas the time to completion of the IV infusion in the Egypt study was 28 minutes [13]. Rates of labor induction and augmentation also varied between studies. In the Egypt study, women who were pre-exposed to uterotonics in labor were excluded from participating in order to evaluate the impact of route of prophylactic oxytocin after delivery on an unexposed uterus [13]. In contrast, over half the sample enrolled in the Ireland trial had their labor induced. In fact, a subgroup analysis of these women showed an even larger reduction in the rate of severe PPH associated with IV bolus, compared to IM injection (Odds Ratio of 0.34 95% CI 0.16–0.72) [12]. Thus, both studies affirmed reductions in blood loss associated with IV routes, compared to IM injection, among women exposed and unexposed to uterotonics prior to the third stage of labor. A meta-analysis of the Adnan and Charles studies together with our findings shows that IV routes, compared to IM injection, were associated with a 23% reduction in the rate of PPH (i.e. ≥500ml) and a 41% reduction in the rate of severe PPH (i.e. ≥1000ml) (Fig 3 and Fig 4).

Despite cumulative evidence favoring IV routes over IM injection for administration of oxytocin to prevent excessive blood loss after childbirth, other factors including provider skill-levels, available resources, and women's preferences, warrant consideration when deciding which route to use. For instance, if an IV line is already in place prior to delivery, use of the same route for administering prophylaxis would be easiest and may reduce the subsequent need for additional uterotonics, based on the findings from this trial. Conversely, if there is no IV line established, IM administration is likely the most efficient way to administer oxytocin safely after delivery of the baby. Of note, in this study, there were no problems reported when administering the study medicine via IM injection; however, two problems did occur during

IV infusion. Overall, difficulties with administration of oxytocin were infrequent but may be more common in other delivery environments with less skilled providers. Importantly, many guidelines on prevention of PPH only recommend the IM route of administration for oxytocin during the third stage of labor [6–8]. Yet, the inclusion of both routes in guidelines would allow for greater flexibility in the clinical care offered to women.

An important study limitation is that the acceptability to women of each route was impossible to evaluate. Due to the fact that the study was placebo-controlled, and each participant received an IM injection and had an IV line in place to administer, it was not possible to ask women to rate their experience with one route over the other—an area that deserves further exploration. Based on the findings from this study and others [12–13], it would be prudent to know whether the clinical advantages of administering prophylactic oxytocin via IV infusion should supersede women's preferences for a less medicalized experience during childbirth. An additional study limitation is that over one-third of the PPH cases diagnosed in our study were due to trauma from episiotomy, which may affect the generalizability of our findings. When designing this study, the underlying assumption was that approximately 90% of PPH cases would be attributable to uterine atony [2,18]. Thus, our sample of atonic PPH, the specific type of PPH that can be prevented by oxytocin during the third stage of labor, might not have been adequate to detect real differences. A larger sample size might have helped compensate for this issue and future studies should modify their assumptions accordingly.

## Conclusions

This study shows that IV infusion and IM injection routes of oxytocin administration did not significantly alter final blood loss outcomes or PPH incidence among participants. Nevertheless, the two routes appear to have resulted in different bleeding patterns immediately after childbirth, which likely triggered the provision of additional uterotonics to those in the IM group. These women appeared to bleed faster and have higher SI values soon after delivery, compared to women who received an infusion of prophylactic oxytocin. Greater awareness of the different clinical effects of IM and IV routes on early bleeding patterns and associated clinical signs may result in more effective tailoring of care during the immediate postpartum period. Importantly though, improvement to prophylactic regimens may have minimal impact on outcomes that matter most to women. More attention is needed to address other clinical practices that may be contributing to an increased bleeding and rates of intervention, including the performance of episiotomy, which was one of the main causes of PPH in this study. Furthermore, efforts to minimize bleeding after childbirth must prioritize access to high quality oxytocin and its proper cold chain storage.

## Supporting information

**S1 Checklist. CONSORT checklist.**
(DOC)

**S1 Fig. Median change in SI from pre-delivery to each 15-minute time interval during the first hour postpartum for PPH cases and non-PPH cases by study group.** *Among the PPH cases, the median change in SI from pre-delivery to 15-minutes postpartum was higher in the IM group (median Δ 0.20, IQR 0.08, 0.33), compared to the IV infusion group (median Δ 0.13 IQR -0.02, 0.22; p = 0.048); change in SI between study groups were comparable at all other time intervals (p>0.05). ^Among the non-PPH cases, the median changes in SIs were not statistically different between study groups at any time postpartum (p>0.05).
(TIF)

**S2 Fig. Proportion of women with SI $\geq$ 0.9 at each 15-minute time interval during the first hour postpartum by study group.**
(TIF)

**S1 Protocol. Research protocol (Original in Spanish).**
(DOC)

**S2 Protocol. Research protocol (English translation).**
(DOC)

## Acknowledgments

We thank the women who participated in the study and the hospital staff who attended to them and collected the necessary data for this study. We are grateful to the hospital staff, medical residents, and administrators for their support of the study, including from Hospital Vidal: Alejandra Gomez, Griselda Abreo, Alberto Cardozo, and Maria Teresa De Sagastizabal. We are also very grateful for the logistical and administrative support to the study provided by the team at the Centro Rosarino de Estudios Perinatales (CREP), specifically to Hugo Gamerro.

## Author Contributions

**Conceptualization:** Jill Durocher, Ilana G. Dzuba, Beverly Winikoff.

**Data curation:** Elba Mirta Morales, Jesus Daniel Aguirre, Roxanne Martin, Jesica Esquivel.

**Formal analysis:** Jill Durocher.

**Funding acquisition:** Beverly Winikoff.

**Investigation:** Jill Durocher, Ilana G. Dzuba, Guillermo Carroli, Beverly Winikoff.

**Methodology:** Jill Durocher, Ilana G. Dzuba, Guillermo Carroli, Beverly Winikoff.

**Project administration:** Elba Mirta Morales, Jesus Daniel Aguirre, Roxanne Martin, Jesica Esquivel, Berenise Carroli.

**Resources:** Roxanne Martin, Berenise Carroli.

**Supervision:** Jill Durocher, Ilana G. Dzuba, Guillermo Carroli, Elba Mirta Morales, Jesus Daniel Aguirre.

**Writing – original draft:** Jill Durocher.

**Writing – review & editing:** Ilana G. Dzuba, Guillermo Carroli, Elba Mirta Morales, Jesus Daniel Aguirre, Roxanne Martin, Jesica Esquivel, Berenise Carroli, Beverly Winikoff.

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
