## [Decision Letter · Decision Letter 0]

23 Jul 2019

PONE-D-19-16212

Does route matter? Impact of route of oxytocin administration on postpartum bleeding: A double-blind, randomized controlled trial

PLOS ONE

Dear Ms. Durocher,

Thank you for submitting your manuscript to PLOS ONE. After careful consideration, we feel that it has merit but does not fully meet PLOS ONE’s publication criteria as it currently stands. Therefore, we invite you to submit a revised version of the manuscript that addresses the points raised during the review process.

We would appreciate receiving your revised manuscript by 08/18/2019. To enhance the reproducibility of your results, we recommend that if applicable you deposit your laboratory protocols in protocols.io, where a protocol can be assigned its own identifier (DOI) such that it can be cited independently in the future. For instructions see: http://journals.plos.org/plosone/s/submission-guidelines#loc-laboratory-protocols

We look forward to receiving your revised manuscript.

Kind regards,

Patrick Rozenberg, MD

Academic Editor

PLOS ONE

Additional Editor Comments (if provided):

Although the topic is not original, the manuscript is technically sound and well written

Please, carefully consider the comments to improve this manuscript

Journal Requirements:

2)  We note that you have indicated that data from this study are available upon request. PLOS only allows data to be available upon request if there are legal or ethical restrictions on sharing data publicly. For information on unacceptable data access restrictions, please see http://journals.plos.org/plosone/s/data-availability#loc-unacceptable-data-access-restrictions.

3) Please include captions for your Supporting Information files at the end of your manuscript, and update any in-text citations to match accordingly. Please see our Supporting Information guidelines for more information: http://journals.plos.org/plosone/s/supporting-information.

Reviewers' comments:

Reviewer's Responses to Questions

**Comments to the Author**

1. Is the manuscript technically sound, and do the data support the conclusions?

Reviewer #1: Yes

Reviewer #2: Yes

Reviewer #3: Yes

2. Has the statistical analysis been performed appropriately and rigorously? 

Reviewer #1: Yes

Reviewer #2: Yes

Reviewer #3: Yes

3. Have the authors made all data underlying the findings in their manuscript fully available?

Reviewer #1: No

Reviewer #2: Yes

Reviewer #3: Yes

4. Is the manuscript presented in an intelligible fashion and written in standard English?

Reviewer #1: Yes

Reviewer #2: Yes

Reviewer #3: Yes

5. Review Comments to the Author

Reviewer #1: This manuscript reports the impact of route of oxytocin administration on postpartum bleeding based on a double-blind, randomized trial. The following are my minor comments.

Table 3, IV group, the n and % should be exchanged for PPH diagnosed. Please indicate if the three excluded women are different from others on baseline characteristics.

Line 191, according to the table, should women (35) be women (53)? Please indicate which group the patients belong for those who received ergonovine.

Fig 2. For SI, it will be more informative to report and compare the average change (with SD) from pre-delivery or median change (with IQR) at each time point and report sample size for each subgroup. Authors may also consider mixed effects model to analyze the trend of SI.

Reviewer #2: In this manuscript, the authors compare intramuscular versus intravenous infusion prophylactic oxytocin for the third stage of labor.

The manuscript is written in standard English. The study objective clearly defined and corresponds to a relevant topic due to a lack of literature on the subject (see Cochrane Oladapo OT 2018: Three studies with 1306 women). However, the study is monocentric and with a sample size not permitting significant results.

Title and Keywords are clear, accurate and matching.

The abstract is accurate and complete. Nevertheless, it seems to me important to specify "IV infusion" in the method paragraph to prevent any potential confusion with the "IV bolus" route. This proposal could be usefully repeated throughout the manuscript.

Regarding the Method section:

Random sequence generation is well described. The blinding of participants and staff seems to be well respected. Research ethics are described (local ethics committee, registered with clinical trials.gov).

However, I have a few comments/questions for the authors.

- The sample size calculation is based on a strong, very optimistic hypothesis: “we hypothesized that administration of oxytocin via IV infusion would result in a 50% lower rate of PPH than IM administration”. This could explain the sample size and the non-significant outcomes.

- Confounders and their management are not described, they should be detailed in statistical analysis

- It would be easier to read the method section if structured according to CONSORT Reporting Guidelines.

- Oxytocin dose should be specified in the International Unit. It is specified "1ml" in the method section.

Regarding the Results section:

- It would seem that the numbers (n) and percentages have been inverted in tables 2 and 3: sometimes it is the number that is in brackets (for example for "uterine massage": "38.1(91)" instead of "91 (38.1)")

- Adverse effects are not described in result section.

- The results describing post-hoc analysis: PPH by uterine atony (l.219-229) do not respond to the research objective described in the method.

Discussion and conclusions are justified by the results. However, study limitations like sample calculation could be also discussed.

Reviewer #3: This randomized study tried to assess the best route of delivery of oxytocin to prevent PPH.

The background is well described and there is a lack of data regarding this question that may be of importance in countries where women do not have intra venous infusion during the third stage of labor.

The study is a well conducted double blind randomized study including 480 women delivering vaginally.

Since Adnan et al Published a larger well conducted randomized trial in 2018, these data are however less “new”.

The main concern is the number of subjects and the hypothesis used to calculate the number of subjects. As there was no real data regarding oxytocin route and that current recommendations do not provide a preferred route of administration, assuming that IV would reduce from 50% the rate of PPH is a very strong hypothesis: the use of oxytocin reduce by 50% the rate of PPH…

I would suppress the post hoc analysis conducted on a selected group of women that can be biased and do not add substantially to the paper.

There are only two or three well randomized trial comparing IV and IM. A metaanalysis of these studies as a last table would really add something to this paper.

In the discussion:

You cannot insist on the contradiction between your results and those previously published (adnan et al) as you results are actually very close : you just have a lack of power and this is what you write in the next chapter of your discussion.

Minor comments:

A detailed flow chart would be useful.

There are some inversion on the tables between n and % (able 2 line 1 , table 3 line 5 …)

6. PLOS authors have the option to publish the peer review history of their article (what does this mean?). If published, this will include your full peer review and any attached files.

Reviewer #1: No

Reviewer #2: No

Reviewer #3: No

---

## [Author Response · Author response to Decision Letter 0]

15 Aug 2019

Responses below to Editor Comments and Reviewers’ Comments

Additional Editor Comments (if provided):

Although the topic is not original, the manuscript is technically sound and well written

Please, carefully consider the comments to improve this manuscript

Response: Thank you for your thorough review of our paper. We believe that the manuscript is significantly improved and are grateful to you and the reviewers who took the time to carefully read it. 

Journal Requirements:

1) Please ensure that your manuscript meets PLOS ONE's style requirements, including those for file naming. The PLOS ONE style templates can be found at http://www.journals.plos.org/plosone/s/file?id=wjVg/PLOSOne_formatting_sample_main_body.pdf and http://www.journals.plos.org/plosone/s/file?id=ba62/PLOSOne_formatting_sample_title_authors_affiliations.pdf

Response: We have improved the formatting of our manuscript, tables, and figures according to the guidance that was shared. 

2) We note that you have indicated that data from this study are available upon request. PLOS only allows data to be available upon request if there are legal or ethical restrictions on sharing data publicly. For information on unacceptable data access restrictions, please see http://journals.plos.org/plosone/s/data-availability#loc-unacceptable-data-access-restrictions.

Response: We are planning to post the dataset to Harvard’s dataverse as soon as the paper is accepted for publication. We will happily provide the DOI at that time. 

3) Please include captions for your Supporting Information files at the end of your manuscript, and update any in-text citations to match accordingly. Please see our Supporting Information guidelines for more information: http://journals.plos.org/plosone/s/supporting-information.

Response: We have included captions for the Supporting Information files at the end of our manuscript. 

Reviewer #1: 

This manuscript reports the impact of route of oxytocin administration on postpartum bleeding based on a double-blind, randomized trial. The following are my minor comments.

Table 3, IV group, the n and % should be exchanged for PPH diagnosed. Please indicate if the three excluded women are different from others on baseline characteristics.

Response: Thank you for noticing this error. We have corrected the n (%) for the IV group for PPH diagnosed in Table 3 and have corrected similar errors in Table 2. 

Regarding the three women excluded from the analysis of postpartum blood loss outcomes, we have expanded the footnote to explain the specific reasons why their blood loss could not be measured for each case (see Lines 211-213). We also have confirmed that their baseline characteristics were not different from other cases included in the analysis. Lastly, we have revised the last two boxes of the Consort Flow Chart to reflect this information regarding the three excluded cases from the primary analysis. 

Line 191, according to the table, should women (35) be women (53)? Please indicate which group the patients belong for those who received ergonovine.

Response: Thank you for asking this question. We have shifted the data from the footnote to the table (see Table 3) and list the Ns of those given additional IV oxytocin and/or ergonovine. We have modified the footnote to clarify that more than one uterotonic could be given, which occurred in only 5 cases (see Line 216). We hope the information is now more clearly presented in the table. In total, 3 women in the IV group and 5 women in the IM group were given ergonovine as an additional uterotonic. 

Fig 2. For SI, it will be more informative to report and compare the average change (with SD) from pre-delivery or median change (with IQR) at each time point and report sample size for each subgroup. Authors may also consider mixed effects model to analyze the trend of SI.

Response: We have modified Fig 2 to include the sample size for each subgroup. We have opted to keep Fig 2 in the manuscript because it provides a useful overview of the SI trends for non-PPH cases and PPH cases, which support recently published findings on the topic, but are still not widely known among all audiences. We also believe that this analysis provides useful insight into the hemodynamic patterns that likely influenced providers’ decisions to offer additional uterotonics. Our finding of more frequent use of additional uterotonics in the IM group than in the IV infusion group, was the main motivation for exploring SI trends by study group among the PPH and non-PPH cohorts. 

We also conducted the analysis suggested by the reviewer to look at the change in SI from before delivery to each time interval measured postpartum. We have added this figure as a supplement to the manuscript. Please see S1 Fig, which is referenced in the Results section in Lines 225-227. While we agree that this new figure provides useful information, we believe that the absolute SI values, as opposed to the delta SI values, were more likely to have influenced provider practices to offer additional care. For one, the baseline SI value may not have been known to the delivery attendant when caring for the woman during the first hour after childbirth. Also, the study team and providers were familiar with new information coming from other studies on the relationship between SI and PPH that showed SI ≥ 0.9 to be a potentially useful marker of PPH and could help aid in determining the need for treatment. Indeed, some of the previous studies on SI (Le Bas et al 2015, Nathan et al 2016, and El Ayadi et al 2016) were included as references in our study protocol to justify data collection on SI and postpartum blood loss. After giving further consideration to all of the above, we have added an additional figure to our manuscript as supplemental to show the proportion of women who had SI ≥ 0.9 by study arm at each time interval postpartum (see S2 Fig and also Lines 234-35 in the Results section). 

We thank the reviewer for suggesting that we consider a mixed effects model to conduct an overall trend analysis of SI. While we think this suggestion is interesting, it would require a deeper dive into the data that is outside the scope and aims of this paper. It is also worth mentioning that we are in the process of preparing a separate manuscript that analyzes the SI findings from this study along with data on SI from other studies to enable a more complete analysis with a larger sample. 

Reviewer #2: 

In this manuscript, the authors compare intramuscular versus intravenous infusion prophylactic oxytocin for the third stage of labor. The manuscript is written in standard English. The study objective clearly defined and corresponds to a relevant topic due to a lack of literature on the subject (see Cochrane Oladapo OT 2018: Three studies with 1306 women). However, the study is monocentric and with a sample size not permitting significant results.

Response: Thank you for this feedback. Although our study’s sample size is small in comparison to recent trials conducted by Charles et al (2019) and Adnan et al (2018), we believe that these findings fill an important gap in the evidence and call attention to the need for greater consideration of the role that route plays in the delivery of oxytocin to reduce postpartum bleeding after childbirth. Our study is the only double-blind RCT to date that compares a prophylactic dose of oxytocin given via IV infusion compared to IM injection. The Adnan trial is the only other double-blind study which tested the bolus route versus IM injection. Although our study findings were not statistically significant, which was likely due to the small size, we believe they provide important information that helps support the findings from the recent larger trials conducted by Adnan et al and Charles et al.

Title and Keywords are clear, accurate and matching. The abstract is accurate and complete. Nevertheless, it seems to me important to specify "IV infusion" in the method paragraph to prevent any potential confusion with the "IV bolus" route. This proposal could be usefully repeated throughout the manuscript.

Response: Thanks you for this suggestion. We have specified “IV infusion” in the methods section of the abstract (see Line 24). We have taken the Reviewer’s suggestion to specify IV infusion throughout the paper where relevant. 

Regarding the Method section: Random sequence generation is well described. The blinding of participants and staff seems to be well respected. Research ethics are described (local ethics committee, registered with clinical trials.gov). However, I have a few comments/questions for the authors. - The sample size calculation is based on a strong, very optimistic hypothesis: “we hypothesized that administration of oxytocin via IV infusion would result in a 50% lower rate of PPH than IM administration”. This could explain the sample size and the non-significant outcomes.

Response: We thank the reviewer for sharing these important observations about our study design. We agree that our sample size calculation was optimistic. We based our hypothesis on results from previous research, including those from an earlier RCT (un-blinded) that randomized participants after vaginal delivery to receive either 10 IU oxytocin via IV infusion or IM injection. This study documented a statistically significant reduction in measured blood loss ≥ 500ml (IV infusion: 9% (15/161) and IM: 20% (32/161); RR 0.47 95% CI 0.25-0.85) and helped to inform our hypothesis for our present study. The results from this prior research can be found online in a published conference abstract from the FIGO World Congress XX held in Rome in October 2012. https://www.sciencedirect.com/science/article/pii/S0020729212606377 We have modified our Methods section (see Lines 137-149) to provide further detail about this prior study and its findings, which provided the basis for the development of our study’s hypothesis. 

Upon further reflection, we could have selected a smaller difference, but that choice might have been criticized for being arbitrary and not clinically significant. Related to the latter point, we specifically aimed to test a large difference between the two routes of administration, so if proven, this difference would signify a clear clinical benefit of one route over the other and one that would supersede programmatic advantages associated with the different routes. We also took into consideration that the study’s primary outcome (blood loss ≥ 500ml) and the fact that it does not necessarily represent an outcome of clinical consequence for all women. For instance, many women tolerate well this level of blood loss after childbirth. And while this outcome is standardly used in PPH prevention trials (Meher S BJOG 2018) and is referred to in international guideline to define PPH, a reduction of its occurrence may be viewed as clinically unimportant if the effect is small-to-moderate. In sum, we believed that a large difference might warrant a change in practices for this prophylactic intervention, but a small-to-moderate difference may not, given that the two routes have different practical advantages in different settings. 

- Confounders and their management are not described, they should be detailed in statistical analysis

Response: We thank the reviewer for raising this question. As we did not find any significant imbalance between the IM and IV study groups with regard to baseline characteristics at enrollment or rates of other AMTSL practices during the third stage of labor, we did not adjust for any confounders in our analysis. 

However, we did complete several sensitivity analyses to explore if our main study outcomes were different among particular subgroups of women (similar to the analyses presented in Table 5 of the published study by Adnan and colleagues). We ran these analyses (post-hoc) to assess outcomes for women who were primigravida; women whose labor was induced; women whose labor was induced/augmented; women who had episiotomy performed; women who had a pre-delivery Hb level < 11 g/dL. None of these analyses produced results that would modify our study conclusions. Also, it is difficult to draw any firm conclusions due the small sample size included in these subgroup analyses. We are happy to share these results with the reviewer below for his/her reference in this response document; however, we do not think they belong in our paper. 

We also ran logistic regression to explore if any of the aforementioned variables has a confounding effect on our study’s primary outcomes. Our main findings remain unchanged after adjusting for potential confounders, including labor induction, primigravida, and episiotomy. 

Summary of sensitivity analyses completed:

 IV infusion IM injection P value

No induction/augment. (n=186) (n=193) 

 Blood loss ≥ 500mL 18.8% (35) 20.2% (39) P=0.733

 Blood loss ≥ 1000mL 5.4% (10) 5.2% (10) P=0.932

Labor induced (n=16) (n=25) 

 Blood loss ≥ 500mL 18.8% (3) 44.0% (11) P=0.096

 Blood loss ≥ 1000mL 12.5% (2) 20.0% (5) P=0.534

Labor induced/augmented (n=52) (n=46) 

 Blood loss ≥ 500mL 26.9% (14) 39.1% (18) P=0.198

 Blood loss ≥ 1000mL 7.7% (4) 17.4% (8) P=0.144

Primigravida (n=93) (n=90) 

 Blood loss ≥ 500mL 30.1% (28) 34.4% (31) P=0.530

 Blood loss ≥ 1000mL 9.7% (9) 14.4% (13) P=0.322

Pre-delivery Hb<11 g/dL (n=61) (n=68) 

 Blood loss ≥ 500mL 24.6% (15) 29.4% (20) P=0.539

 Blood loss ≥ 1000mL 9.8% (6) 8.8% (6) P=0.843

No episiotomy performed (n=135) (n=130) 

 Blood loss ≥ 500mL 11.1% (15) 14.6% (19) P=0.394

 Blood loss ≥ 1000mL 3.0% (4) 2.3% (3) P=0.739

Episiotomy performed (n=103) (n=109) 

 Blood loss ≥ 500mL 33.0% (34) 34.9% (38) P=0.776

 Blood loss ≥ 1000mL 9.7% (10) 13.8% (15) P=0.360

- It would be easier to read the method section if structured according to CONSORT Reporting Guidelines.

Response: We have added subheadings to the Methods section to help orient the reader. We would welcome confirmation and further instructions from the journal if these types of sub-headings are allowed. 

- Oxytocin dose should be specified in the International Unit. It is specified "1ml" in the method section.

Response: Thank you for sharing this observation. We have changed it to say 10 IU in the Methods section (see Line 96).

Regarding the Results section:

- It would seem that the numbers (n) and percentages have been inverted in tables 2 and 3: sometimes it is the number that is in brackets (for example for "uterine massage": "38.1(91)" instead of "91 (38.1)")

Response: Thank you for noticing these errors, which have been corrected in Tables 2 and 3. 

- Adverse effects are not described in result section.

Response: Thank you for asking this question. Providers were asked to document any adverse effect that occurs since administration of the prophylactic regimen and none were reported. We have added at sentence to the Results section confirming that there were no adverse effects reported in our study (see Lines 264-265). 

- The results describing post-hoc analysis: PPH by uterine atony (l.219-229) do not respond to the research objective described in the method.

Response: Thank you for this feedback. In light of your comment and feedback from another reviewer suggesting that we omit this post-hoc analysis, we have decided to remove this analysis from the Results section (removed Lines 252-263). We agree that it does not add substantially to the paper and main findings. 

Discussion and conclusions are justified by the results. However, study limitations like sample calculation could be also discussed.

Response: We thank the reviewer for this comment. We have added more detail to the Methods section to explain the rationale for our study hypothesis and sample size calculation (see Lines 137-149), which we hope is helpful. While we agree that our study assumption was ambitious, we do not necessarily consider it a limitation of our study. The real limitation, we believe, is that our sample was diluted by PPH caused by episiotomy, for which prophylactic oxytocin is likely to have no effect, and that we incorrectly assumed that the majority of PPH cases would be attributable to uterine atony. We agree that a larger sample size may have helped to compensate for this issue and that future studies should modify their assumptions accordingly.

Reviewer #3: 

This randomized study tried to assess the best route of delivery of oxytocin to prevent PPH.

The background is well described and there is a lack of data regarding this question that may be of importance in countries where women do not have intra venous infusion during the third stage of labor. The study is a well conducted double blind randomized study including 480 women delivering vaginally. Since Adnan et al Published a larger well conducted randomized trial in 2018, these data are however less “new”.

Response: We believe our study findings and design are complementary to the large trial conducted by Adnan et al. Our study is the only double-blind RCT to date that compares a prophylactic dose of oxytocin given via IV infusion compared to IM injection. The Adnan trial is the only other double-blind study which tested its IV bolus administration versus IM injection. Together we believe these results fill an important void in the evidence on the effect of IV routes of oxytocin administration on postpartum blood loss. 

The main concern is the number of subjects and the hypothesis used to calculate the number of subjects. As there was no real data regarding oxytocin route and that current recommendations do not provide a preferred route of administration, assuming that IV would reduce from 50% the rate of PPH is a very strong hypothesis: the use of oxytocin reduce by 50% the rate of PPH…

Response: In response to a similar comment by Reviewer #2, we have modified our Methods section (see Lines 137-149) to provide further detail on the research findings that served as the basis for the development of our study’s hypothesis. 

I would suppress the post hoc analysis conducted on a selected group of women that can be biased and do not add substantially to the paper.

Response: We appreciate this feedback Reviewer #2 had a similar comment and we agree that it does not add substantially to the paper. See Lines 252-263 in the Results section that has been removed in tracked changes. Please note that we have retained one sentence in the Discussion section that reports on the timing of PPH diagnosis among women with atonic PPH.

There are only two or three well randomized trial comparing IV and IM. A meta-analysis of these studies as a last table would really add something to this paper.

Response: We thank the reviewer for this suggestion. We agree that a meta-analysis of the three recent studies that our discussed in our paper would fit well in our manuscript, including our study, the Adnan trial and the 3-arm trial by Charles et al. Please note that we also had considered if the other three studies from the 2018 Cochrane review should be included in this meta-analysis, but identified numerous differences between the studies with regard to their study design and how the outcomes were reported, resulting in the decision to only include the most recent studies. Of note, none of the studies in the last Cochrane review were double-blinded. Please see Lines 321-28 where we report on the meta-analysis findings and have added the new figures (Fig 3 and 4).

In the discussion:

You cannot insist on the contradiction between your results and those previously published (adnan et al) as your results are actually very close : you just have a lack of power and this is what you write in the next chapter of your discussion.

Response: Thank you for this observation. We agree that our results were very close to previously published results by Adnan et al and Charles et al and have removed the sentence in our Discussion section that describes our findings as contradictory (see Lines 296-298). We agree that our study lacks power and while our sample size could have been larger, the lack of power in our study might be due in part to the unexpectedly high rate of traumatic cases of PPH due to episiotomy, which may have prevented us from seeing any real differences between the study arms. See Lines 351-352 in our Discussion where we raise this issue as a study limitation.

Minor comments:

A detailed flow chart would be useful.

Response: We are not certain what level of detail is being requested or suggested by this comment. Our manuscript was accompanied by Figure 1: Consort Flow Chart that describes the enrollment process and total number of cases analyzed. In response to feedback received by other reviewers, we have added more detail to the Consort Flow Chart. It would be helpful to know if the Reviewer has had a chance to see this flow chart and if the information presented is sufficient. We kindly request further instructions on what type of flow chart and additional detail would be useful and if it is still recommended after reviewing our revisions to the manuscript, including the addition of subheadings to the Methods section. 

There are some inversion on the tables between n and % (able 2 line 1 , table 3 line 5 …)

Response: Thank you for noticing these errors, which have been corrected in Tables 2 and 3.

---

## [Editor Report · Decision Letter 1]

12 Sep 2019

Does route matter? Impact of route of oxytocin administration on postpartum bleeding: A double-blind, randomized controlled trial

PONE-D-19-16212R1

Dear Dr. Durocher,

We are pleased to inform you that your manuscript has been judged scientifically suitable for publication and will be formally accepted for publication once it complies with all outstanding technical requirements.

With kind regards,

Patrick Rozenberg, MD

Academic Editor

PLOS ONE
---

## [Editor Report · Acceptance letter]

20 Sep 2019

PONE-D-19-16212R1 

Does route matter? Impact of route of oxytocin administration on postpartum bleeding: A double-blind, randomized controlled trial 

Dear Dr. Durocher:

I am pleased to inform you that your manuscript has been deemed suitable for publication in PLOS ONE. Congratulations! Your manuscript is now with our production department. 

With kind regards,

on behalf of

Professor Patrick Rozenberg 

Academic Editor

PLOS ONE